# Recent Advances of Tannase: Production, Characterization, Purification, and Application in the Tea Industry

**DOI:** 10.3390/foods14010079

**Published:** 2024-12-31

**Authors:** Zhanhui Tang, Liyu Shi, Shuang Liang, Junfeng Yin, Wenjiang Dong, Chun Zou, Yongquan Xu

**Affiliations:** 1College of Biological and Environmental Sciences, Zhejiang Wanli University, Ningbo 315100, China; t15355402324@163.com (Z.T.); shiliyu@zwu.edu.cn (L.S.); 2Tea Research Institute, Chinese Academy of Agricultural Sciences, Key Laboratory of Biology, Ministry of Agriculture and Rural Affairs, 9 South Meiling Road, Hangzhou 310008, China; liangshuang@tricaas.com (S.L.); yinjf@tricaas.com (J.Y.); 3Spice and Beverage Research Institute, Chinese Academy of Tropical Agricultural Sciences, Wanning 571533, China; dongwenjiang.123@catas.cn; 4National Engineering Research Center for Tea Processing, Hangzhou 310008, China

**Keywords:** Tannase, fermentation, enzymatic characterization, purification, application

## Abstract

Tannase, as a type of tannin−degrading enzyme, can catalyze the hydrolysis of ester and depside bonds in gallotannins, thereby releasing gallic acid and glucose. Based on this reaction mechanism, Tannase can effectively improve the problems of bitter taste, weak aroma, and tea cheese in tea infusion, and is therefore widely used in the tea industry. However, due to high production costs, difficulties in purification and recovery, and insufficient understanding of Tannase properties, the large−scale application of Tannase is severely limited. Therefore, the sources of Tannase and the effects of fermentation temperature, pH, stirring speed, time, carbon, and nitrogen sources on the preparation of Tannase are described in this study. The advantages and disadvantages of various methods for measuring Tannase activity and their enzymatic characterization are summarized, and the concentration and purification methods of Tannase are emphasized. Finally, the application of Tannase to reduce the formation of tea precipitate, enhance antioxidant capacity, increase the extraction rate of active ingredients, and improve the flavor of the tea infusion is described. This study systematically reviews the production, characterization, purification, and application of Tannase to provide a reference for further research and application of Tannase.

## 1. Introduction

Tannins, mainly divided into hydrolyzed and condensed tannins, are a class of water−soluble polyphenols with self−protection ability, with a molecular weight of about 500–3000 Da [1]. Tannins are widely distributed in various parts of higher plants, such as fruit, seeds, leaves, roots, bark, etc., which are produced by plants to protect themselves from animals and microorganisms. The chemical structures of hydrolyzed and condensed tannins, as well as tannin−rich plants, are shown in Figure 1. Tannins can combine with proteins, polysaccharides, cellulose, and digestive enzymes to form complexes [2]. However, tannins can cause many problems with tea. For example, during the storage process of tea infusion, due to the large number of phenolic groups in tannins, it is easy to form complexes and precipitate with other components such as proteins, which not only affects the appearance of the product but also makes it less appealing to consumers [3]. Tannins are also effective chelators of metal ions, which can affect the absorption of these essential elements [4]. In addition, reducing the tannin content in tea is important for minimizing the formation of precipitates in tea infusion. Tannin−degradation methods are divided into physical, chemical, and enzymatic degradation. Due to the high costs, risks, and significant nutritional losses associated with physical and chemical methods, these methods cannot be applied on a large scale. The enzymatic degradation method is widely used because plants, animals, and microorganisms can produce Tannase in large quantities. Among these enzymatic degradation methods, Tannase is the most studied tannin−degrading enzyme [5]. 

Tannase, the full name of tannin acyl hydrolase, is usually an extracellular enzyme. Some microorganisms synthesize it in the presence of tannic acid and other inducers, which is classified as an inducible enzyme. Tannase can catalyze the hydrolysis of ester and depside bonds in gallotannins, releasing gallic acid and glucose [6]. Tannase has found widespread use in the food, industrial, pharmaceutical, and animal feed sectors. As its applications expand, the demand for Tannase is increasingly outpacing its supply. However, due to the high production costs, the difficulty in purification and recovery, and the lack of understanding of Tannase characterization, the large−scale application of Tannase is severely limited [7]. This study reviews the sources, activity determination methods, characterization, purification methods, and applications of Tannase in the tea industry, providing a reference for researchers to make better use of Tannase in the industry.

**Figure 1 foods-14-00079-f001:**
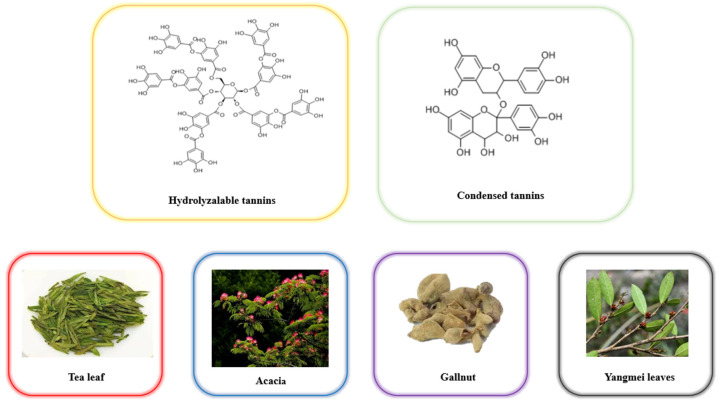
Chemical structure of hydrolyzed and condensed tannins and tannin−rich plants.

## 2. Production of Tannase

### 2.1. Source of Tannase

Currently, most studies use microorganisms to produce Tannase, mainly including bacteria and fungi. The Tannase−producing strains are summarized in Table 1.

Bacteria have the advantage of more efficient degradation of tannins and easier regulation at the genetic level [8]. Currently, bacteria that have been confirmed to have Tannase production capacity include *Bacillus subtilis* [9], *Enterobacter cloacae* [10], *Klebsiella pneumoniae* [8], *Pseudomonas aeruginosa* [11], *Lactobacillus pentosus* [12], *Staphylococcus lugdunensis* [13], et al. Some fungi have the advantage of rapid growth and easy separation from the fermentation solution, and they can tolerate tannic acid concentrations up to 20% without adverse effects on fungal growth and Tannase production [14]. There are more than 100 common fungi that can produce Tannase, most of which belong to *Aspergillus* and *Penicillium* [15]. These include *Aspergillus niger* [16], *A. awamori* [17], *Penicillium verrucosum* [18], *Rhodosporidium diobovatum* [19], et al.

The direct production of Tannase by wild microorganisms has disadvantages, such as low expression levels, high costs, and complex purification processes. Therefore, recombinant microorganisms are constructed through molecular biology technology to improve the catalytic efficiency and yield of Tannase [1,20]. At present, studies on recombinant expression of fungi reported include *A. oryzae* [21], *A. niger* [22], *R. diobovatum* [19], etc. Zhong et al. [21] cloned a Tannase gene from *A. oryzae*, expressed it in *Pichia Pastoris*, and measured the activity of the recombinant enzyme. The study found that the recombinant Tannase increased the yield by 3.5 times and the carbohydrate content by 8.3% compared with solid fermentation. Pan et al. [19] cloned and expressed the Tannase gene *TANRD*. The results showed that the Tannase activity was slightly higher than that of the original strain and exhibited higher activity at 25−60 °C and pH 2.5−6.5. Bacterial Tannases usually have only one subunit, which makes them easier to clone, overexpress, and purify than fungi, which consist of multiple subunits [13]. At present, these studies on recombinant expression are reported: *L. pentosus* [12], *S. lugdunensis* [13], *Streptococcus gallolyticus* [23], etc. Kanpiengjai et al. [12] isolated two strains of lactic acid bacteria from fermented tea for isolation, cloning, and expression. They found that one strain, LpTanBA−7, was considered to be an alkaline thermophilic Tannase with activity and stability in various organic solvents. Zhao et al. [24] screened a Tannase gene with promising enzymatic properties using a deep learning approach (DLKcat), and the gene was synthesized and transferred into *Escherichia coli* to achieve heterologous expression of a novel Tannase. The results demonstrated that the recombinant Tannase had a high affinity for propyl gallate, indicating its strong potential for industrial applications. Jiménez et al. [23] extracted DNA from *St. gallolyticus* UCN34 to amplify the Tannase gene *tanSg1* using specific primers. *Escherichia coli* BL21 was employed as the host cell to express the *tanSg1* gene. The resulting Tannase demonstrated stability across a wide temperature range and exhibited high specificity for gallate and protocatechuate, making it a valuable enzyme for industrial applications.

**Table 1 foods-14-00079-t001:** Tannase−producing strains.

Microorganism	Source	Strain	Expression Mode	GenBank:	Reference
Fungus	Pods of *Caesalpinia spinosa*	*A. allahabadi*	Natural expression	−	[25]
	Sea water	*A. awamori* BTMFW032	Natural expression	GQ337057	[17]
	−	*A. niger*	Natural expression	−	[16]
	Marine sediment	*A. nomius* GWA 5	Natural expression	−	[26]
	−	*A. tubingensis* CICC 2651	Natural expression	−	[27]
	Coffee by−products	*P. verrucosum* CFR 303	Natural expression	KM213866	[18]
	Aqueous areca extract	*Geotrichum cucujoidarum*	Natural expression	MW538971	[28]
	−	*A. oryzae*	Recombinant expression	−	[21]
	−	*A. niger* PTCC 5012	Recombinant expression	AY727901	[22]
	Mangrove−derived	*R. diobovatum* Q95	Recombinant expression	MW173231	[19]
	−	*Pseudoduganella albidiflava*	Recombinant expression	WP_131147687.1	[24]
Bacteria	−	*B. subtilis* PAB2	Natural expression	HM853662.1	[9]
	Gastrointestinal tract of goat	*B. cereus* M1GT	Natural expression	KX033490	[29]
	Compost sample	*E. cloacae* MTCC 9125	Natural expression	−	[10]
	Feces, soil, fruits, and vegetable garbage	*K. pneumoniae* KP715242	Natural expression	KP715242	[8]
	Soil sample	*Ps. aeruginosa* IIIB 8914	Natural expression	−	[11]
	Gut contents of fish	*Raoultella ornithinolytica*	Natural expression	−	[30]
	Ruminant gastrointestinal tract	*Lachnospiraceae bacterium*	Recombinant expression	MBQ6323131.1	[31]
	Fermented tea leaves	*L. pentosus strain* BA−7	Recombinant expression	MH169607	[12]
	Fermented tea leaves	*L. pentosus strain* QA1−5	Recombinant expression	MH169608	[12]
	−	*S. lugdunensis* MTCC 3614	Recombinant expression	KU882097.1	[13]
	−	*St. gallolyticus* UCN34	Recombinant expression	YP_003431024	[23]

### 2.2. Preparation of Tannase

Currently, numerous studies report the isolation and screening of Tannase−producing strains from natural sources, with most following a sequence of steps that include sample collection, enrichment culture, primary screening, and re−screening [32,33,34,35]. This study describes the Tannase−preparation process in detail (Figure 2).

**Figure 2 foods-14-00079-f002:**
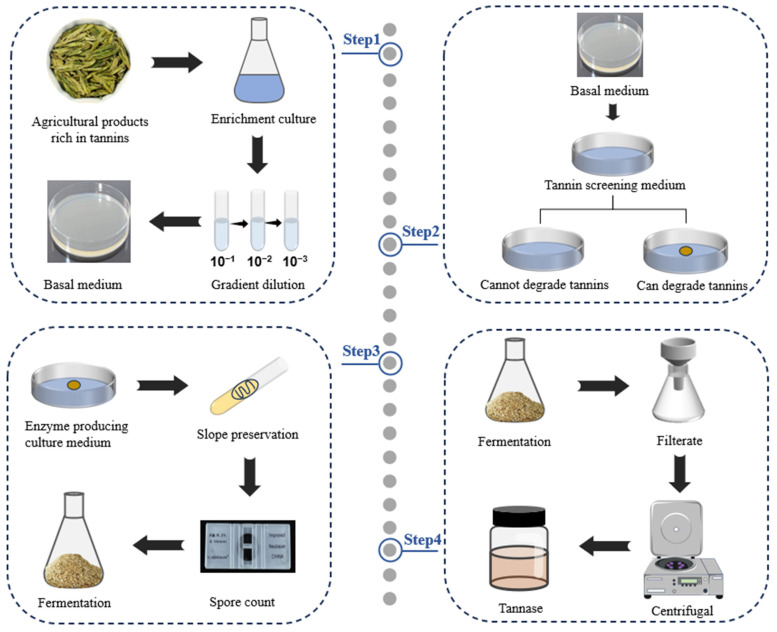
Tannase preparation process.

First, tannin−rich samples (such as tea leaves, gallnut, soil, etc.) were selected and placed in a sterilized Erlenmeyer flask containing distilled water. The samples were then enriched and cultured on a shaking incubator. Afterward, the culture solution was serially diluted, and the diluted microbial suspension was plated onto the appropriate base medium. Once single colonies appeared, the inoculation loop was used to transfer the single colonies to agar slants for further cultivation. The cultures were stored at 4 °C.

Since Tannase specifically degrades tannic acid to produce gallic acid, tannic acid is used as the sole carbon source when preparing the primary screening medium, with an indicator added. If the strain grows on the medium and forms a hydrolysis zone, it is considered that the strain can produce Tannase to degrade tannic acid. The strains isolated in the first step were inoculated into the primary screening medium and cultured in an incubator. The growth of each colony was observed, along with the formation of any discoloration zones. The diameters of the discoloration zone (D) and the colony (d) were measured, and the D/d ratio was used as the preliminary screening index to assess the enzyme−production capacity of the strain. Single colonies with larger D/d ratios were selected, transferred to agar slant culture medium, cultured in an incubator, and finally stored at 4 °C.

The primary screening medium can remove most strains that do not meet the requirements, but the size of the clear or colored zones may not accurately reflect the level of enzyme production [36]. Therefore, further verification through double screening is necessary to precisely determine the enzyme−production capacity of each strain. For this purpose, a fermentation enzyme−production medium was prepared using tannic acid or tannin−rich agricultural products as the substrate. The spores from the agar slant culture medium were washed into a sterile Erlenmeyer flask with normal saline, and the spore suspension was diluted to an appropriate concentration. This suspension was subsequently used for the fermentation of Tannase. After fermentation, buffer was added to the Erlenmeyer flask containing the solid fermentation medium, and crude Tannase was obtained by filtration and centrifugation. The enzyme activity of Tannase was determined using the standard enzyme activity assay, with the enzyme activity level serving as the rescreening index. Afolayan et al. [37] selected tannin−rich soil for enrichment culture. Each sample was inoculated onto Potato Dextrose Agar (PDA) plates via gradient dilution and cultured at 28 °C for 120 h. After the culture, the microorganisms were purified several times. The strains were added to the primary screening medium and fermented at 28 °C for 96 h. After fermentation, the tannic acid hydrolysis zone was detected by using a 1% FeCl_3_ solution as an indicator. Sorghum pomace was taken as solid substrate, and spore suspension was inoculated into the medium, mixed evenly, and cultured at 28 °C for 120 h. After fermentation, 250 mL of sodium acetate buffer (0.1 M, pH 5.5) was added, and the mixture was shaken. Then, it was filtered through gauze and filter paper. Finally, the filtrate was centrifuged at 5000 rpm for 20 min, and the supernatant was collected as the crude Tannase solution. Thakur et al. [38] isolated and screened tannin−producing enzyme strains from tannin−rich orchards, tea leaves, botanical gardens, and other soils. The soil samples were enriched and cultured, then plated on PDA medium containing 0.5% tannic acid for fungal screening. The primary screening medium, with tannic acid as the sole carbon source, was prepared. The microbial suspension was applied to the primary screening medium, and fermentation was carried out at 28 ± 2 °C. Tannase−producing strains were screened based on the hydrolysis cycle (D/d), and enzyme activity was measured. For solid−state fermentation (SSF), pine needles were used as the substrate. Solid substrates were compacted, moistened with an appropriate amount of Czapdox minimal medium (CMM) broth, inoculated with spore suspension, and cultured at 30 ± 2 °C. After fermentation, a suitable buffer was added for enzyme extraction. Finally, the supernatant obtained by centrifugation was collected as the crude Tannase solution.

### 2.3. Fermentation Conditions of Tannase

In the fermentation process, the efficiency of microbial Tannase production mainly depends on temperature, pH, stirring speed, fermentation time, carbon source, and nitrogen source. This article summarizes these aspects to optimize the cultural conditions and understand the effects of different carbon and nitrogen sources on microbial growth and Tannase production (Table 2).

#### 2.3.1. Temperature

The optimal temperature range for microbial fermentation to produce Tannase is 25−45 °C [10,11,18,19,30]. Fermentation at low temperatures may reduce Tannase production because low temperatures decrease the efficiency of substrate transport into the cells. At higher temperatures, metabolic pathways may undergo thermal denaturation, and the increased energy demand for maintaining cell growth could further reduce Tannase production [39]. Pan et al. [19] screened Tannase−producing strains at low temperature, ground mangrove plant samples, evenly coated them on Yeast Extract−Peptone−Tannic acid (YPT) plates, and then transferred single colonies to YPT liquid medium for fermentation at 25 °C for 48 h. Shakir et al. [30] cultured *Ra. ornithinolytica* at temperatures of 37, 40, and 45 °C. The results indicated that Tannase production peaked at 45 °C. Peña−Lucio et al. [40] used an exploratory experimental design (Hunter and Hunter) to study the effect of temperature (25, 30, and 35 °C) on the Tannase fermentation process. The results showed that temperature at 30 °C significantly affected Tannase yield, reaching 246.82 U/g.

#### 2.3.2. pH

Most Tannase−producing strains ferment in the pH range of 3.0−7.0 [8,11,19,30,41]. The pH of the production medium plays a vital role in metabolite production, mainly because pH affects the ionization state of acidic and essential amino acids [30]. To improve the yield of Tannase, Pan et al. [19] cultured the Tannase−producing strain in YPT medium with a pH of 3.0, and found that the Tannase activity was as high as 26.4 U/mL when the tannic acid concentration was 15 g/L. Saad et al. [42] studied the effects of different pH values (4.0, 5.0, 6.0, 7.0, and 8.0) on the Tannase production efficiency of *A. glaucus*. The results showed that in the fermentation process, the enzyme−production efficiency was highest when the pH was 5. Shakir et al. [30] optimized the SSF of *Ra. ornithinolytica* and found that Tannase yield was highest at pH 7.0 through single−factor optimization, while Tannase activity peaked at pH 5.0 using the response surface method.

#### 2.3.3. Mixing Speed

According to statistics, during the fermentation process of Tannase production, the stirring speed is generally controlled between 90 and 200 rpm [16,17,18,25]. During fermentation, the level of dissolved oxygen in the medium is directly proportional to the stirring speed, and variations in stirring speed impact the extent of oxygen mixing in the shaker. Thus, for Tannase−producing strains, the efficient oxygen transport and adequate mixing are critical for effectively utilizing nutrients [43]. Beena et al. [17] used submerged fermentation (SMF) at a stirring speed of 90 rpm, and the final specific activity of the Tannase was 2761.89 IU/mg. Natarajan et al. [44] studied the effects of different stirring speeds (50–200 rpm) on Tannase yield under constant conditions and found that the highest Tannase activity occurred at a stirring speed of 125 rpm. However, the Tannase activity decreased rapidly when the stirring speed exceeded 200 rpm, and stirring speeds below 150 rpm will result in insufficient broth mixing at the later stage of growth.

#### 2.3.4. Incubation Time

As reported, microorganisms generally produce Tannase within 12−168 h of fermentation [10,18,45,46]. According to previous studies, strains typically start to produce Tannase after 12 h. However, excessive fermentation time can also lead to a decrease in Tannase production, which is due to the accumulation of gallic acid leading to a more acidic medium [43]. To study the effect of different fermentation times on Tannase production, Belur et al. [45] used the response surface method of full factor center complex design to optimize the medium. The results showed that *B. massiliensis* reached the highest Tannase activity of 9.65 U/L after fermentation for only 12 h, which was 93.8% higher than before optimization. Lal et al. [46] took samples at 30 °C for 3–14 d and measured Tannase activity every 24 h. The results showed that the Tannase activity peaked at 162.3 U/mL on the seventh day of fermentation. Alfarhan et al. [47] studied the effect of incubation time (1, 2, 3, and 4 d) on Tannase yield at 37 °C. The results showed that the highest Tannase activity was 29.3 ± 1.9 U/mL on day 3.

#### 2.3.5. Carbon and Nitrogen Sources

Tannase is an inducible enzyme that requires tannin−rich agricultural byproducts such as tannic acid to be produced in the presence of inducers and carbon sources. Tannins play a crucial role in microbial growth, Tannase induction, and metabolite production. Moreover, Tannase production is also influenced by other carbon sources, such as glucose, alpha−lactose, and mannose [6]. Lal et al. [46] added mannose, galactose, glycerol, and ribose with concentrations of 0.5−2% to replace tannin in the medium, respectively. The results showed that all carbon sources reduced Tannase yield, as Tannase is an inducible enzyme. Wu et al. [48] explored the effects of different carbon sources on microbial growth and Tannase production by adding lactose, starch, maltose, sucrose, glucose, and tannin (6%) to the medium. The study found that among these carbon sources, glucose had the most significant effect, increasing biomass accumulation by 107% and significantly enhancing Tannase activity. Paranthaman et al. [49] studied the effects of different tannic acid concentrations (0.02, 0.04, 0.06, 0.08, and 0.1%) on Tannase activity, and the results showed that when tannic acid concentration was 0.06%, the total Tannase activity reached the highest level of 27.8 U/gm/min.

Nitrogen sources also play a crucial role in microbial growth and secondary metabolism. Wu et al. [48] added (NH_4_)_2_NO_3_, NH_4_Cl, (NH_4_)_2_SO_4_, urea, beef extract, peptone, yeast extract, and corn infusion into the medium as nitrogen sources and found that (NH_4_)_2_SO_4_ was the most suitable nitrogen source for Tannase production among all the nitrogen sources. In addition to urea, all organic nitrogen sources also had a promoting effect on Tannase production. Lal et al. [46] replaced sodium nitrate with equal amounts of different nitrogen sources, including inorganic nitrogen sources such as ammonium sulfate, ammonium chloride, ammonium nitrate, ammonium dihydrogen phosphate, sodium nitrate, and potassium nitrate, as well as organic nitrogen sources such as peptone, yeast extract, beef extract, urea, and casein. The results showed that in addition to sodium nitrate, ammonium nitrate exhibited the highest activity, while urea showed the lowest. Belur et al. [45] adopted the response surface method to optimize the medium to study the influence of nitrogen source fermentation in a shaker. The results showed that trypsin could increase the Tannase activity to 9.65 U/L, which was 93.8% higher than that before optimization.

Currently, most studies focus on the influence of single carbon and nitrogen sources on fermentation conditions. These studies maintain other conditions unchanged while investigating the effects of a single carbon or nitrogen source. However, these methods are time−consuming and cumbersome, and the final data cannot reflect the interactions between carbon and nitrogen sources. Mohapatra et al. [50] used Taguchi orthogonal experimental design to optimize the medium for Tannase production by examining six critical factors, including carbon and nitrogen sources. They found that the best combination of carbon and nitrogen sources was tannic acid and NH_4_Cl, and the Tannase activity increased by 2.18 times. Wu et al. [48] first used single−factor experiments to show that glucose and yeast extract can significantly increase the accumulation of microbial biomass, and tannin and NH_4_NO_3_ can significantly increase Tannase yield. Then, the response surface method was used to optimize these four factors simultaneously. The results showed that the Tannase activity increased 6.36 times when the supplemental level of tannin was 7.49%, glucose was 8.11%, (NH_4_)_2_SO_4_ was 9.26%, and yeast extract was 2.25%. Belur et al. [45] found that tannic acid is a known Tannase inducer. However, it is easy to form a complex with proteins in the medium, which makes subsequent purification difficult. Therefore, this study used gallic acid as the inducer, and the "protein−free tannin complex" medium was used to optimize its full−factor center complex design. The results showed that the optimized medium comprised 38 g/L lactose, 50 g/L trypsin, and 2.8 g/L gallic acid, and the Tannase activity was up to 9.65 U/L.

**Table 2 foods-14-00079-t002:** Fermentation conditions of Tannase.

Microorganism	Fermentation Method	Temperature (°C)	pH	Agitation (rpm)	Incubation Time (h)	Carbon Source	Nitrogen Source	Enzyme Activity	Reference
*K. pneumoniae* KP715242	SMF	37	5.5	150	48	Tannic acid	NH_4_Cl	0.065 U/mL	[8]
*Ra. ornithinolytica*	SSF	45	5	150	24	Tannic acid	Yeast extract	157.04 U/mL	[30]
*B. subtilis* PAB2	SSF	35	−	−	72	Tamarind seed	NH_4_Cl	73.44 U/gds	[9]
*Ps. aeruginosa* IIIB 8914	SMF	37	7	200	24	Amla and keekar leaves	NH_4_NO_3_	13.65 U/mL (amla), 12.9 U/mL (keekar leaves)	[11]
*A. awamori* BTMFW032	SMF	30	5	90	48	Tannic acid	NaNONaNO_3_	−	[17]
*A. allahabadi*	SMF	30	5	180	40	Tannic acid	−	0.68 U/mL	[25]
*A. glaucus*	SSF	30	5	−	120	black tea waste	NaNONaNO_3_	46.71 U/mL	[42]
*A. niger*	SSF	30	5	−	72	−	NaNONaNO_3_	1.86 U/gds	[16]
*A. niger*	SSF	37	5.5	−	120	Sorghum pomace	NH_4_NO_3_	76.63 U/mL	[37]
*A. flavus* TF−8	SSF	28	5	−	72	Shorea robusta deoiled seed cake	NaNONaNO_3_	599 U/gds	[51]
*A. tubingensis* CICC 2651	SSF	25	6.5	−	118	Dried tea stalk powder	NH_4_Cl	84.24 U/gds	[27]
*A. niger* JMU−TS528	SSF	30	−	−	96	Tea stem powder	(NH_4_)_2_SO_4_	−	[52]
*P. sp.* CFR303	SSF	30	5	−	96	Coffee pulp	NH_4_NO_3_	116 U/gds	[18]
*R. diobovatum* Q95	SMF	25	3	−	48	Tannic acid	(NH_4_)_2_SO_4_	26.4 U/mL	[19]
*E. cloacae* MTCC 9125	SMF	37	−	150	48	Tannic acid	NaNONaNO_3_	0.96 U/mL	[10]
*A. niger*	SMF	30	5	−	168	Tannic acid	NaNO_3_	201.23 U/mL	[46]
*L. plantarum* MTCC 1407	SMF	30	6	125	60	−	−	9.29 U/mL	[44]
*B. licheniformis* KBR6	SMF	40	5	−	−	Tannic acid	NH_4_Cl	0.36 U/mL	[50]
*A. fumigatus*	SMF	37	5	120	24	Tannic acid	NH_4_Cl	27.07 U/mL	[36]
*A. niger*	SMF	37	5	120	24	Tannic acid	−	36.34 U/mL	[36]
*B. massiliensis*	SMF	30	5	150	12	Gallic acid	Tryptose	9.65 U/L	[45]

SMF: submerged fermentation, SSF: solid−state fermentation.

## 3. Characterization of Tannase

At present, various methods are available for determining Tannase activity, including the colorimetric method, titration method, high−performance liquid chromatography, and gas chromatography. Each method has its advantages and disadvantages, and this study will provide a detailed introduction to them (shown in Table 3).

### 3.1. Determination Methods of Tannase Activity

#### 3.1.1. Colorimetric Method

This method [53] is based on the action of Tannase on methyl gallate to release gallic acid. The gallic acid then reacts with rhodanine to form a chromotropic substance, which exhibits maximum absorption at 520 nm. The main steps of this method involve mixing Tannase and substrate for incubation, adding methanolic rhodanine solution to stop the reaction, adding KOH solution to ensure the best color−rendering effect, and finally adding water to measure the absorbance at 520 nm. Chaitanyakumar et al. [13] first incubated 50 μL of Tannase solution and 100 μL of methyl gallate substrate at 37 °C for 5 min, then added 300 μL of methanolic rhodanine solution (0.667%, *w*/*v*) to terminate the reaction, and the mixture was incubated for an additional 3 min. Following this, 100 μL of KOH (0.5 M) was added. Water was added so that the final volume was 2 mL, and the absorbance was measured at 520 nm. The method was controlled by thermally denaturing the enzyme. Gaikaiwari et al. [25] incubated 250 μL of Tannase solution with methyl gallate (dissolved in 0.01 M citric acid buffer, pH 5.0) at 30 °C for 15 min. Then, 300 μL of methanolic rhodanine (0.667%, *w*/*v*) was added to react with the product, gallic acid, and the mixture was incubated for an additional 5 min. Afterward, 200 μL of KOH (0.5 M) and 4 mL of distilled water were added, and the absorbance was recorded at 520 nm. The Tannase activity unit in this method is defined as the hydrolysis of methyl gallate by 1 mL of Tannase solution within 1 min to produce 1 μmol of gallic acid under the experimental conditions. The method offers the advantages of short duration, convenience, and good repeatability. However, due to methyl gallate not being a specific substrate for Tannase, this method may be interfered with by tannic acid in the medium during color formation [54]. 

The colorimetric method established by Mondal is based on the fact that tannic acid easily binds to and precipitates proteins. At the same time, SDS−triethanolamine is an alkaline detergent, which can destroy the tannin–protein complex, and then FeCl_3_ is added to react with the phenolic group of tannin to form brown through nucleophilic reaction [54]. The main steps of this method involve mixing Tannase and tannic acid for a specified period, then adding bovine serum albumin (BSA) solution to terminate the reaction and precipitate the remaining tannic acid. The resulting precipitate is then dissolved in SDS−triethanolamine by centrifugation, and FeCl_3_ is added for color development, and finally, the absorbance is measured at 530 nm. Selwal et al. [11] mixed 0.3 mL of tannic acid (dissolved in 0.2 M sodium acetate, 0.5% *w*/*v*, pH 5.5) with 0.1 mL of Tannase and incubated the mixture at 30 °C for 60 min. Then, 3 mL of BSA solution (1 mg/mL) was added to precipitate the remaining tannic acid and stop the reaction. After centrifugation (5000 g, 10 min), the obtained precipitate was dissolved in SDS−triethanolamine (1% *w*/*v* SDS dissolved in 5% *v*/*v* triethanolamine), and finally 1 mL of FeCl_3_ (0.01 M FeCl_3_ dissolved in 0.01 M HCl) was added and held for 15 min to stabilize the color. The absorbance was measured at 530 nm, and a thermal denaturing enzyme controlled the method. Kumar et al. [55] incubated 0.2 mL of Tannase solution with 0.3 mL of tannic acid (dissolved in 0.2 M acetic acid buffer, 1.0% *w*/*v*, pH 5.5) at 40 °C for 40 min. Then, 3 mL of BSA (1 mg/mL) was added to stop the reaction, and the solution was centrifuged (10,000 rpm, 10 min) to precipitate the remaining tannic acid. The precipitate was dissolved in 3 mL of SDS−triethanolamine (1.0% *w*/*v*, triethanolamine), and 1 mL of FeCl_3_ (0.13 M) was added and held for 15 min to stabilize the color. The absorbance was measured at 530 nm, and a thermal denaturing enzyme controlled the method. The Tannase activity unit of this method is defined as the amount of enzyme that can hydrolyze 1 mM of the substrate tannic acid in 1 mL of enzyme solution within 1 min under experimental conditions. This method has the advantages of good repeatability and low cost, but it is relatively cumbersome and time−consuming compared with the above spectrophotometric method.

#### 3.1.2. Titration Method

The titration method produces free gallic acid through the action of Tannase on the substrate tannic acid, titrates the free gallic acid with a base, and determines the Tannase activity by calculating the reduction of tannic acid in the substrate [56]. Haslam et al. [56] hydrolyzed part of the substrate solution with Tannase, and the gallic acid produced was continuously titrated with 0.01 M NaOH solution so that the pH of the reaction medium was always maintained at 6.0. The enzyme activity was determined by recording the consumption of NaOH over time. The unit of Tannase activity defined in this method is the amount of enzyme that can release 1 mmol of carboxyl groups in 1 min in 1 mL of enzyme solution under experimental conditions [57]. This method has the advantages of being economical, fast, and simple. However, the stability and accuracy of the determination result are not good because of the interference of tannic acid color, buffers, and other factors [58].

#### 3.1.3. Chromatography

Sarıözlü et al. [59] established a selective and highly accurate method for determining Tannase activity by high−performance liquid chromatography (HPLC). The method is to measure the amount of gallic acid produced in the system after the reaction, using the product with a specific absorption peak at 254 nm for detection. The repeatability of the method is improved by adding internal standards.

Sarıözlü et al. [59] established acetaminophen as the internal standard and used the C18 column for separation. The mobile phase consisted of formic acid aqueous solution (1%) and methanol (85:15; *v*/*v*), the flow rate was 1 mL/min. A total of 10 μL of sample was injected, and the signal was detected at 254 nm. The Tannase activity unit of the method is defined as the Tannase activity of hydrolyzing 1 μM tannate ester bond per minute under experimental conditions, referred to as one unit. Although this method is more accurate, sensitive, and specific in determining Tannase activity, it has the disadvantages of expensive instruments and high operator requirements. Therefore, few studies have used this method to determine enzyme activity.

Gas chromatography (GC) [60] is to enter the mixture to be separated into the gas chromatography instrument through the injection port and then pass the mobile phase and sample mixture through the column filled with a fixed phase. Different compounds are separated according to their adsorption and desorption rates on the fixed phase, so as to stay in the column for different times. Finally, the signals of different compounds are detected and analyzed by the detector to determine the type and content of compounds in the sample. Jean et al. [61] used gas chromatography for determination and introduced the separation and determination methods of soluble Tannase and fixed Tannase, respectively. Although this method is more accurate, sensitive, and specific to determine Tannase activity, it has the disadvantages of expensive instruments and high requirements for operators, so there are few studies using this method to determine enzyme activity [58]. 

At present, there are many methods to determine Tannase activity, and each method has its own advantages and disadvantages, so it is necessary to understand the limitations of each method according to its actual situation and then make a choice. 

**Table 3 foods-14-00079-t003:** Methods for the determination of Tannase activity.

Substrate	Terminators and Color Developing Agents	Wavelength	Reference
Tannic acid	BSA and FeCl_3_	530 nm	[11]
Tannic acid	BSA and FeCl_3_	530 nm	[55]
Tannic acid	BSA and FeCl_3_	530 nm	[54]
Tannic acid	3, 5−dinitrosalicylic	540 nm	[62]
Tannic acid	Ethanol	310 nm	[62]
Methyl gallate	KOH and rhodanine	520 nm	[13]
Methyl gallate	KOH and rhodanine	520 nm	[25]
Methyl gallate	KOH and rhodanine	520 nm	[12]
Methyl gallate	KOH and rhodanine	520 nm	[26]
Methyl gallate	KOH and rhodanine	520 nm	[53]
Methyl gallate	KOH and rhodanine	520 nm	[63]
Methyl gallate	NaHCO_3_	450 nm	[64]

### 3.2. Enzymatic Properties of Tannase

At present, the physicochemical characterization of Tannase has been a significant research focus in its study. The source and fermentation conditions of Tannase are the main reasons for the differences in its enzymatic characteristics [7]. The characterization of Tannase, including molecular weight, pH, temperature, metal ions, and organic solvents, is summarized in Table 4.

#### 3.2.1. Molecular Weight

The molecular weight of Tannase, as currently reported, generally ranges from 31 to 320 kDa [10,12,65,66]. Beniwal et al. [10] purified *E. cloacae* MTCC 9125 by ultrafiltration and ion exchange chromatography and then analyzed by Sodium Dodecyl Sulfate Polyacrylamide Gel Electrophoresis (SDS−PAGE). The results showed that the molecular weight of the Tannase was 31 kDa. Böer et al. [65] respectively determined the Tannase derived from natural and recombinant strains and found that the two strains were composed of four identical subunits to form a 320 kDa tetramer. Pan et al. [19] cloned and expressed the Tannase−coding gene *TANRD*. The purified TANRD was analyzed by SDS−PAGE, revealing a molecular weight of approximately 75.1 kDa.

The molecular weight of Tannase is also closely related to the number and types of subunits. According to previous studies, the molecular weight of bacterial Tannase is generally between 31−90 kDa, while fungal Tannase has a larger molecular weight, usually between 45−320 kDa [67]. Hatamoto et al. [68] found that the natural Tannase of *A. oryzae* consisted of a 300 kDa polymer composed of four pairs of two types of subunits via disulfide bonds. Riul et al. [69] found that the Tannase from *A. phoenicis* had a molecular weight of 218 kDa and two subunits of 120 kDa and 93 kDa, respectively. Qiu et al. [70] reported a Tannase derived from *P. herquei*, which was found to be a monomeric protein without a disulfide binding subunit, with a molecular weight of 72 kDa. Iwamoto et al. [71] expressed *tanLpl* in *L. plantarum* ATCC 14917 and analyzed the purified recombinant enzyme. The results showed that the single protein band was about 50 kDa, and the Tannase did not contain intermolecular disulfide bonds, indicating that it can function as a monomer. Jana et al. [9] analyzed the pure Tannase through SDS−PAGE and found that the electrophoretic profile had only one band, and the molecular weight of the enzyme was about 52 kDa.

#### 3.2.2. Optimum pH and pH Stability

The effect of pH changes on Tannase activity is significant, and Tannase usually has a wide pH stability range. Under normal circumstances, the optimum pH and stable range of Tannase are 2.0−9.0 and 3.0−10.0, respectively [9,12,17,23]. Beena et al. [17] isolated *A. awamori* from seawater and cultured it to produce extracellular Tannase. The study found that the enzyme had two optimal pH values, 2.0 and 8.0, respectively. However, the activity of this Tannase could remain stable for 24 h only at pH 2.0, while its was unstable at other pH values, indicating that it was an acid−resistant Tannase. Kanpiengjai et al. [12] found that LpTanBA−7 and LpTanQA1−5 from *L. pentosus* were active under alkaline conditions, and the optimal pH values were 8.0 and 9.0, respectively. Singh et al. [51] investigated the optimal pH for Tannase derived from *A. flavus* TF−8 by measuring enzyme activity under different pH conditions. The results indicated that Tannase exhibited the highest activity at pH 6 and remained stable within the pH range of 4−7.

Tannase derived from different microorganisms exhibit varying optimal pH values and stability ranges. It has been reported that the optimal pH of fungal Tannase is mostly around 6.0, while the optimal pH of bacteria is between 7.0−9.0 [19]. Böer et al. [65] cloned the Tannase gene *ATAN1* and expressed the recombinant enzyme. The results showed that the optimal pH of the recombinant enzyme was 6.0. Sharma et al. [66] reported that the optimal pH of Tannase derived from *P. variable* was 5.0, and the stability range was 3.0−8.0. Iwamoto et al. [71] tested the purified recombinant Tannase under different buffer conditions for pH stability and found that the optimal pH of the enzyme was 8.0 and the stability range was 6.0−9.0. Jiménez et al. [23] measured the pH stability range of the recombinant Tannase TanSg1 and found that the enzyme had high stability in 6.0–8.0.

#### 3.2.3. Optimum Temperature and Temperature Stability

In general, the optimal temperature and temperature stability of microbial Tannases are between 30–60 °C and 30–70 °C, respectively [6]. It has been demonstrated by previous studies that Tannase achieves its maximum reaction rate at the optimal temperature. When the temperature exceeds this value, the reaction rate decreases or even becomes inactive due to the impact of thermal stability [72]. Beena et al. [17] studied the effect of temperature on Tannase activity. The results showed that the enzyme was active in the range of 10–70 °C, but the enzyme was only stable at 30 °C for 24 h, maintaining about 60% activity. Ramos et al. [73] studied the influence range of different temperatures (20–65 °C) on enzyme activity at pH 5. The results showed that the Tannase from *A*. *niger* was stable at 25–65 °C, and the affinity of Tannase for methyl gallate was higher at 45 °C. Osipov et al. [74] studied the optimal temperature and thermal stability of recombinant Tannase TAN2 by heat treatment at 50, 60, and 70 °C for 10−180 min. The results showed that TAN2 exhibited optimal activity at 45 °C, with no more than a 50% decrease in activity within the temperature range of 30–65 °C.

It has been reported that fungal Tannases exhibit higher activity and stability than bacterial−derived enzymes under various temperature conditions [19]. Pan et al. [19] investigated the enzymatic properties of the recombinant Tannase TanRd at different temperatures and found that the enzyme activity peaked at 40 °C and maintained 70% activity at 25–60 °C. Sharma et al. [66] reported an immobilized enzyme with a functional temperature range of 25–80 °C. This Tannase has ideal properties and stability under extreme temperatures. Beena et al. [17] cultured Tannase in the range of 5–100 °C, and the results showed that the enzyme maintained about 60% activity at 30 °C for 24 h, and the highest activity was 72% at 70 °C for 1 h.

#### 3.2.4. Metal Ions

Among various metal ions, Mg^2+^, Ca^2+^, and Mn^2+^ can activate Tannase in most cases, while heavy metal ions (Hg^2+^, Cd^2+^, Ni^2+^, and Cs^2+^) can inhibit Tannase activity [10,12,67,75]. Chhokar et al. [75] studied the effects of different metal ions on Tannase at the concentration of 1 mM, and the results showed that Mg^2+^, Mn^2+^, Ca^2+^, Na^+^, and K^+^ improved the Tannase activity to different degrees, and Cu^2+^, Fe^3+^, and Co^2+^ inhibited the Tannase activity. To study the effect of metal ions on Tannase production, Lal et al. [46] added different concentrations (1–20 mM) of metal ions (Zn^2+^, Mo^6+^, Al^3+^, Mn^2+^, Cu^+3^, and Li^3+^) to the medium, and the results showed that all metal ions had an inhibitory effect on Tannase activity. Jana et al. [9] studied the effect of metal ions on *B. subtilis* PAB2 Tannase. The results showed that Mg^2+^ with a concentration of 1 mM could significantly improve the purified Tannase activity. The inhibitory effects of metal ions on Tannase were Fe^3+^, Hg^2+^, Fe^2+^, Ba^2+^, Ca^2+^, Co^2+^, Zn^2+^, Ag^+^, and K^+^. Guan et al. [31] studied the effects of different metal ions (K^+^, Ca^2+^, Co^2+^, Mn^2+^, Mg^2+^, Zn^2+^, and Cu^2+^) on recombinant Tannase TanALb at 1 mM. The results showed that Mg^2+^ and Ca^2+^ increased Tannase activity by 130% and 122%, and Cu^2+^ and Mn^2+^ decreased Tannase activity by 33% and 51%, respectively. K^+^, Co^2+^, and Zn^2+^ had little effect on the enzyme activity.

#### 3.2.5. Organic Solvents

According to previous studies, Tannase activity is inhibited by most organic solvents. Since Tannase’s tolerance to organic solvents is crucial for industrial applications, finding Tannases resistant to organic solvents has become one of the key areas of current research [17,75]. Chhokar et al. [75] found that acetic acid, isoamyl alcohol, chloroform, and isopropyl alcohol inhibited Tannase activity at different concentrations, while butanol and benzene increased enzyme activity. Gonçalves et al. [76] discovered a strain of extracellular Tannase with high tolerance to organic solvents. The results showed that isopropyl alcohol, acetonitrile, and ethanol (50%, *v*/*v*), as well as SDS, Triton X−100, and Ethylenediaminetetraacetic acid (EDTA), could enhance Tannase activity at low concentrations. Chaitanyakumar et al. [13] studied the effect of organic solvents on Tannase activity. The results showed that non−polar solvents (hexane, toluene, and benzene) increased Tannase activity, while polar solvents (methanol, dimethyl sulfoxide (DMSO), isoamyl alcohol, and butanol) inhibited Tannase activity.

**Table 4 foods-14-00079-t004:** Enzymatic properties of Tannase.

Microorganism	Molecular Weight (kDa)	Optimum		Stable Range		Activator	Inhibitor	Reference
		pH	Temperature (°C)	pH	Temperature (°C)			
*E. cloacae* MTCC 9125	31	5.5	50	4.5−5.5	40−80	Mg^2+^, Zn^2+^, and Mn^2+^	Fe^2+^, Ba^2+^, and Cu^2+^	[10]
*La. bacterium*	68	7	50	6.5−7.5	30−55	Mg^2+^ and Ca^2+^	Cu^2+^ and Mn^2+^	[31]
*L. pentosus strain* BA−7	50	8	50	7−9	20−65	Ca^2+^, Mg^2+^, and Ni^2+^	Co^2+^ and Cu^2+^	[12]
*L. pentosus strain* QA1−5	50	9	40	8−10	30−60	Ca^2+^, Mg^2+^, and Ni^2+^	Co^2+^ and Cu^2+^	[12]
*L. plantarum* ATCC 14917	50	8	40	6−9	25−45	−	−	[71]
*B. subtilis* PAB2	52	5	40	3−8	30−70	Mg^2+^ (1 mM)	Fe^3+^, Hg^2+^, Fe^2+^, Ba^2+^, Ca^2+^, Co^2+^, Zn^2+^, Ag^+^, and K^+^	[9]
*K. pneumoniae* KP715242	38.20	5	50	4.5−5.5	30−50	Mg^2+^, Mn^2+^, and Zn^2+^	Methyl gallate, Hg^2+^, xylene, and sodium thioglycolate	[8]
*St. gallolyticus* UCN34	52.98	7	50	6−8	50−70	Ca^2+^	Zn^2+^ and Hg^2+^	[23]
*Arxula adeninivorans*	320	6	40	−	−	−	Phenyl methyl sulfonyl fluoride, Cd^2+^, and Cu^2+^	[65]
*A. awamori* BTMFW032	−	2/8	30	−	10−70	Na^+^ (10 mM), K^+^ (1−10 mM), and F^3+^ (1 mM)	Phenanthroline, phenyl methane sulfonyl fluoride, and sodium deoxycholate	[17]
*A. awamori* MTCC9299	101	5.5	30	5−6	25−70	Mg^2+^, Mn^2+^, Ca^2+^, Na^+^, K^+^, butanol, and benzene	Cu^2+^, Fe^3+^, Co^2+^, acetic acid, isoamyl alcohol, chloroform, isopropyl alcohol, and ethanol	[75]
*A. niger* GH1	−	6	60	2−8	20−65	−	−	[73]
*A. nomius* GWA5	30	6	50	3−9	40−70	Mg^2+^	Zn^2+^, Cd^2+^, Hg^2+^, Pb^2+^, Ba^2+^, and EDTA	[26]
*Emericella nidulans*	302	5	45	4−7	22−50	Zn^2+^, Hg^2+^, Co^2+^, isopropyl alcohol, acetonitrile, ethanol, SDS, and TritonX−100 detergent	Fe^3+^, Al^3+^, Ag^+^, and β−mercaptoethanol	[76]
*P. variable*	310	5	50	3−8	25−80	Sodium choleate and sodium taurocholeate	Phenyl methyl sulphonyl fluoride, N−ethylmaleimide, SDS, Tween−60, and Tween−80	[66]
*R. diobovatum* Q95	75.1	4.5	40	3−8	25−60	Mn^2+^ and Co^2+^ (1 mM)	Cu^2+^, Al^3+^, Ba^2+^, Ca^2+^, Cu^2+^ (10 mM), and 2−hydroxy−1−ethanethio (1 mM)	[19]
*Rhodotorula glutinis* DB2	73	6	40	6−7	0−40	Fe^3+^, Sr^2+^, Na^+^, and Pb^2+^ (1 mM)	Ba^2+^, Ca^2+^, Mg^2+^, Zn^2+^, Hg^+^, Ag^+^, Co^2+^, Fe^2+^, Mn^2+^, Cu^2+^, Cd^2+^, Al^3+^, K^+^, Ni^2+^, and Li^+^ (1 mM)	[77]
*S. lugdunensis* MTCC 3614	66	7	40	4−9	30−40	SDS (1%), Na^+^ (1 mM), hexane, toluene, and benzene	Zn^2+^, Fe^2+^, Fe^3+^, Mn^2+^, β−mercaptoethanol (1%), EDTA (1%), methanol, DMSO, isoamyl alcohol, and butanol	[13]

## 4. Purification Method of Tannase

Although various purification methods are available, the final selection should be based on market demand and environmental considerations. For example, in commercial applications, Tannase requires higher purity to ensure production efficiency and product quality, while in the treatment of waste liquid−related fields, more attention is paid to the cost price, and the cost is reduced as far as possible if the requirements are met [78]. At present, conventional Tannase purification is mainly divided into two steps. The first step involves concentrating the crude enzyme using simple methods such as ammonium sulfate precipitation, dialysis, and ultrafiltration. The second step further purifies the enzyme using techniques like chromatography and extraction. The various purification methods for Tannase are summarized in Table 5.

### 4.1. Ammonium Sulfate Precipitation

In the ammonium sulfate precipitation method [79], the addition of high−concentration salt to the protein solution leads to the exclusion of the hydration layer, the increase of surface tension, and the disruption of the hydration shell on the protein surface, making the protein hydrophobic and forming precipitation. This phenomenon is reversible. When the salt concentration in the protein solution is reduced, the hydration shell of the protein is restored, thus making it hydrophilic, and the protein precipitate can be dissolved again. Since (NH_4_)_2_SO_4_ has a higher solubility than any phosphate, it is the reagent of choice for salting out. Govindarajan et al. [80] gradually mixed the collected crude Tannase solution with ammonium sulfate powder until the concentration reached 50%. After complete precipitation, the protein was precipitated by centrifugation at 10,000× *g* rpm and then dissolved by adding a small amount of sodium phosphate buffer (0.2 M, pH 6.0). Finally, a dialysis membrane (20 kDa) was used to remove the residual ammonium sulfate. The results showed that the recovery rate of Tannase was 87.66%, and the specific activity was 2.43. This method has the advantages of wide application, low cost, high efficiency, and simplicity. However, it can only concentrate crude enzymes, which need to be combined with other purification methods to improve purity, and after concentration, the enzymes are in high concentration of salt, often through ultrafiltration, dialysis, and other methods for desalting treatment [80,81,82,83,84].

### 4.2. Acetone Precipitation

The principle of acetone precipitation is that the addition of acetone lowers the dielectric constant of the aqueous solution, which increases the interaction between oppositely charged groups, disrupting the protein’s hydration shell, and promotes the aggregation and precipitation of protein molecules [79,83]. Farag et al. [26] purified Tannase using ammonium sulfate, ethanol, and acetone precipitation. The results showed that the highest recovery rates of protein and Tannase activity were achieved through precipitation with 75% acetone, at 13.82% and 17.09%, respectively, and the specific activity was 1.59 times that of the crude enzyme solution. Therefore, 75% acetone was selected for the next step of purification. Gaikaiwari et al. [25] added acetone at three times the volume of the sample at −20 °C, precipitated at 4 °C for 3 h to complete precipitation, then centrifuged the sample at 4 °C for recovery (20 min, 10,000 g), and finally dried the precipitation in a vacuum dryer to remove the acetone. The results showed that by acetone precipitation and tangential flow filtration (TFF) concentration, the purification rate of Tannase was 6.76 times, and the yield was 52.44%. This method has the advantages of high separation capacity and simple operation, but organic solvents can denature and deactivate proteins, which need to be removed with subsequent purification methods [83,85].

### 4.3. Dialysis

The dialysis method [86,87] is based on the difference in molecular weight of solutes in solution to achieve purification. During dialysis, small and medium−sized solute molecules are diffused from high−concentration solution to low−concentration solution until equilibrium is reached. This process is driven by the concentration gradient. A semi−permeable membrane with specific pore sizes selectively allows smaller molecules (such as water and salts) to pass through, while preventing larger molecules (such as proteins) from passing. This selective permeability allows dialysis to separate substances based on their molecular sizes. Ammonium sulfate precipitation is a commonly used method for Tannase purification. However, at the end of precipitation, a large amount of ammonium sulfate precipitates along with the protein. Therefore, dialysis is often used for desalting ammonium sulfate precipitation, to obtain an ideal pure enzyme solution. Kumar et al. [8] precipitated crude Tannase using ammonium sulfate, and to desalt the protein, the sample was dialyzed in 0.01 M acetic acid buffer (pH 5.5) for 24 h, during which the buffer was repeatedly changed. Dhiman et al. [87] concentrated the crude enzyme using ammonium sulfate precipitation. After concentration, dialysis to remove salts was carried out in citric acid buffer (0.05 M, pH 5.5) using a dialysis membrane with a molecular weight cutoff of 14,000. This method has the advantages of simple operation and wide application and mild conditions, but it is relatively time−consuming and needs to be combined with other purification methods to improve purity [83,86].

### 4.4. Ultrafiltration

The principle of ultrafiltration [10,83] is based on membranes with pores of varying sizes, which selectively exclude large molecules (such as proteins) while allowing smaller molecules to pass through. When the mixture passes through the ultrafiltration membrane, large protein molecules are retained on the membrane, while solvents, salts, and smaller molecules pass through the pores. Beniwal et al. [10] removed tannic acid from the fermentation solution by ultrafiltration and further concentrated the enzyme. First, the fermentation solution was centrifuged in a 100 kDa ultrafiltration tube at 3000 g for 20 min, and the Tannase activity and total protein of the obtained retentate and permeate were measured. The retention was further enriched with 30 kDa MWCO membrane with the same parameters. The results showed that the purification rate of the enzyme was 1.78 times and the recovery was 37.1%. The method has the advantages of high selectivity, environmental friendliness, and mild conditions, but the membrane fouling can significantly reduce the performance of ultrafiltration, resulting in an increase in cost.

### 4.5. Extraction

#### 4.5.1. Two−Phase Aqueous Extraction

Two−phase aqueous extraction [85] is a commonly used separation technique based on a two−phase system consisting of two immiscible solvents. In this method, the target compounds are initially distributed between the two phases, selectively distributed in one of the phases according to their affinity. By mixing the two phases and waiting for the extract to be transferred from one phase to the other, effective separation is achieved. Ma et al. [88] selected the operation point of the two−water phase system according to the phase diagram, added polyethylene glycol (PEG) reserve solution, sodium citrate solution and Tannase extract into a 10 mL centrifuge tube, supplemented the insufficient part with distilled water, so that the total mass reached 10 g, and fully stirred the mixture in a vortex mixer for 5 min. Centrifugation was performed at room temperature (10 min, 954 × gat) to accelerate the formation of two aqueous phases. The volume of each phase was directly estimated by centrifugal tube scale, and the activity and protein concentration of each phase were determined. The results showed that the purification fold of Tannase activity reached 2.74, and the recovery rate was 77.17%. This method can produce high−purity Tannase with the advantages of high productivity, low cost, minimal equipment requirements, and easy scalability. However, it may lead to environmental pollution [89,90,91].

#### 4.5.2. Reverse Micelle Extraction

The reverse micelle method [92] is a commonly used protein purification method in which surfactants can form micelles in which a hydrophobic part surrounds the protein, keeping it dissolved in an aqueous solution. When appropriate ions or organic solvents are added, these substances can destroy the micellar structure, causing the protein to dissociate from the micelle. The protein can then be separated from the solution by centrifugation or other separation techniques for purification purposes. Gaikaiwari et al. [25] studied different surfactants, different concentrations of cetyltrimethylammonium bromide (CTAB), mixing time, pH of fermentation solution, and the ratio Vaq:VRM (crude Tannase solution: effect of organic phase/reverse micelle) on back extraction. The forward extract was mixed and stirred at a certain temperature and time and then centrifuged at 8000× *g* for 10 min at 25 °C to separate the two phases, and the residual activity of Tannase in the aqueous phase was determined. The organic phase was back extracted at 30 °C, and the reverse extraction solution was NaCl solution (0.5 M in 50 mM citric acid buffer, pH 5.0). The results showed that under optimized conditions, the CTAB−isoctane system was the most suitable, the purification rate was 12.7 times, and the recovery rate was 81.2%. This method has the advantages of fewer steps and shorter time consumption. However, the purification fold of this method is not high [25,83,92,93].

### 4.6. Protein Chromatography Purification

#### 4.6.1. Ion Exchange Chromatography

The principle of ion exchange chromatography [94,95] is based on the electrostatic interaction between the protein and the ion exchange medium. The ion exchange medium is usually a resin or gel containing ion exchange groups that are able to adsorb or repel ions or molecules with opposite charges. Proteins usually have a positive or negative charge and are subject to electrostatic adsorption in ion−exchange media. By changing the ionic strength, pH value, or ionic composition of buffer, the interaction between the protein and ion exchange medium can be adjusted to achieve selective adsorption and elution of protein. In this way, proteins with different charge properties can be separated to achieve purification effect. Beniwal et al. [10] further purified the concentrated Tannase obtained by ultrafiltration using ion exchange chromatography, equilibrated with 0.02 M acetic acid buffer (pH 5.5), and eluted with 0.02 M acetic acid buffer (pH 5.5, 0–0.5 M NaCl) in linear gradient (0.5 mL/min). The results showed that the enzyme’s purification fold was 8.78, with a recovery rate of 33.1%. Selwal et al. [96] purified Tannase using a two−step process involving ammonium sulfate precipitation and ion exchange chromatography. The concentrated crude samples were applied to diethylaminoethyl cellulose (DEAE−cellulose column) (2 × 42 cm), and the Tannase was eluted with 0–1 M NaCl in linear gradient. In the DEAE−cellulose column, the Tannase recovery was 27.6%, the purification fold was 23, and the specific enzyme activity was 1562 U/mg. This method has the advantages of processing a large number of samples, broad applicability, and low cost. However, the samples must be under low−salt conditions. [94,97].

#### 4.6.2. Dextran Gel Chromatography

Dextran gel chromatography [78,83] is a commonly used method for protein purification, which achieves effective separation of proteins based on the molecular size exclusion effect. The porous structure of dextran gels allows larger protein molecules to be hindered and diffuse only along the surface, while smaller molecules can enter the internal pores, causing different−sized protein molecules to move through the gel at different rates, thus achieving separation. By adjusting the buffer conditions, protein components of different sizes can be eluted and collected sequentially to achieve separation. Gonçalves et al. [76] carried out dialysis and freeze−drying of the Tannase concentrate after ion−exchange chromatography, purified it on the Sephacryl S−200 column (1 × 80 cm), and balanced it with 50 mM Tris−HCl buffer (pH 7.5, containing 50 mM NaCl). The samples were eluted at a flow rate of 0.33 mL/min, and analyzed by dialysis, lyophilization, and electrophoresis. The results showed that the enzyme’s purification fold was 61.61, with a recovery rate of 29.95%. Ong et al. [77] equilibrated Sephadex G−200 with citric acid buffer (50 mM, pH 6.0), and eluted the concentrated enzyme solution at a flow rate of 0.17 mL/min until there was no absorbance at 280 nm. The results showed that the recovery rate of the purified enzyme was 5.7%, the specific enzyme activity was 1.12, and the purification fold was 110. This method has the advantages of being fast, repeatable, and requiring low polymer demand, but it cannot handle large molecular proteins and has poor selectivity [83,98,99].

**Table 5 foods-14-00079-t005:** Purification methods of Tannase.

	Purification Method	Purification Fold	Yield (%)	Reference
*E. cloacae* MTCC 9125	Ultrafiltration	1.78	37.1	[10]
*A. awamori* BTMFW032	Ultrafiltration	1.62	55	[17]
*A. allahabadi*	Acetone precipitation	6.76	52.44	[25]
*A. nomius* GWA5	Acetone precipitation	1.59	50.04	[26]
*B. haynesii* SSRY4 MN031245	Ammonium sulfate, dialysis	11.93	60	[87]
*B. cereus* KMS3−1	Ammonium sulfate, dialysis	1	87.66	[80]
*Enterococcus faecalis*	Ammonium sulfate, dialysis	9.7	89.8	[100]
*A. awamori* MTCC9299	Ion exchange chromatography	19.5	13.5	[75]
*A. niger*	Ion exchange chromatography	15.11	20.04	[101]
*A. fumigatus*	Ion exchange chromatography	23	27.6	[96]
*Em. nidulans*	Sephadex S−200	61.61	29.95	[76]
*B. licheniformis*	Sephadex G−75 Gel	3.16	12.4	[78]
*Rh. glutinis* DB2	Sephadex G−200	110	5.7	[77]
*A. ficuum* Gim 3.6	Two aqueous phase extraction	2.74	77.17	[88]
*P. rolfssi* URM 6216	Two aqueous phase extraction	18.18	23.37	[102]
*A. allahabadi*	Reverse micelle extraction	12.7	81.2	[25]

## 5. Application of Tannase in the Tea Industry

Tea is a traditional beverage in China. Due to its remarkable antioxidant, anti−cancer, and neuroprotective properties, tea has become one of the most widely consumed beverages in the world [103]. However, during tea processing, several significant issues arise, such as browning, weak aroma, turbidity, and precipitation, which can seriously affect the color, aroma, and taste of the product. Although many companies have developed various chemical methods to address these issues, these technologies are difficult to promote on a large scale due to their complexity and high cost. Tannase can catalyze the hydrolysis of ester and depside bonds in hydrolyzed tannins or gallates, releasing glucose and gallic acid. The gallic acid produced can compete with phenolic compounds to bind with caffeine and reduce the turbidity of tea beverages, in addition to improving flavor, reducing bitterness, and enhancing antioxidant capacity. It has improved the quality and nutritional value of tea beverages [104]. This chapter introduces the application of Tannase in the tea industry from different aspects. The application of Tannase in tea beverages is shown in Figure 3.

### 5.1. Reduce the Formation of Tea Precipitate

In the process of tea processing, production, and storage, the high−temperature extract of tea, which contains polyphenols, proteins, caffeine, and other substances, interacts with caffeine, resulting in precipitation in the tea extract after cooling. As storage time increases, this phenomenon becomes more pronounced, and the precipitation significantly reduces the appearance of the tea beverage. As a result, consumers’ desire to purchase is reduced [3]. Although the precipitation can be removed by centrifugation, filtration, and other methods, it may lead to the reduction in the effective components of tea and affect its flavor [105]. Tannase can hydrolyze ester catechins in tea infusion, and the product gallic acid can compete with phenolic compounds for caffeine, thus preventing the accumulation of macromolecules and the formation of precipitation during the storage of tea beverages [106]. Wang et al. [3] studied the effect of Tannase on the formation of green tea brewing precipitation. The results showed that a Tannase concentration of 0.5 mg/mL and a hydrolysis time of 120 min were the optimal hydrolysis parameters for reducing sediments. Tannase treatment not only reduced the precipitation of green tea extract (74.63%) but also improved the taste after storage. Lu et al. [106] studied the effects of Tannase on the protein–tannin complex and sensory properties in green tea. The results showed that epigallocatechin gallate (EGCG) and epicatechin gallate (ECG) in green tea were hydrolyzed by Tannase to epigallocatechin (EGC) and epicatechin (EC), respectively, releasing gallic acid. Moreover, the binding ability of polyphenols and protein was reduced, which improved the color of green tea, and the sensory evaluation was better than that of the control group. Aharwar et al. [107] treated black tea and green tea with immobilized Tannase at 60 °C for 20 min, respectively, using samples without enzyme as the control. The results showed that the tannin content and turbidity of black tea and green tea were decreased to some extent after treatment with immobilized Tannase, and the effect of black tea was the most significant.

### 5.2. Enhance the Antioxidant Activity of Tea

When people exercise or have bad living habits, the increase in reactive oxygen species in the body may cause a series of health problems [108]. At present, a large number of studies have proven that drinking more tea plays an important role in human health, with benefits such as anti−diabetes, cancer prevention, and cardiovascular disease protection, with antioxidants being the main reason for these effects. Polyphenols, mainly EGCG, ECG, EGC, and EC, are the primary natural antioxidants in tea [109]. At present, many studies have applied Tannase to tea to enhance its antioxidant effect. ONG et al. [110] used liquid chromatography−mass spectrometry to determine the contents and components of polyphenols in tea before and after Tannase treatment. The results showed that after 30 min of Tannase treatment, EGCG and ECG in green tea extract were basically converted into EGC and EC by Tannase, and the antioxidant activity was increased by 43%. It also exhibited higher DPPH radical scavenging activity. Macedo et al. [111], taking green tea and yerba mate as research subjects, used oxygen radical−absorbance capacity (ORAC) and radical scavenging ability (DPPH) methods to determine the changes in tea extracts before and after Tannase treatment. The results showed that the antioxidant capacity of green tea and mate treated with Tannase was 55% and 43% higher than that of untreated tea, respectively. This may be due to the fact that Tannases can hydrolyze two important substrates in tea extracts (chlorogenic acid from yerba mate and EGCG from green tea), and their hydrolyzed products enhance the antioxidant capacity. Rajendra et al. [112] studied the combined treatment of tyrosinase (1785 U/mL) and Tannase (1 mg/mL) to prepare black tea with high antioxidant activity. The results showed that a higher concentration of tyrosinase reduced the antioxidant capacity of black tea, but it was still higher than that of commercial black tea and tea treated only with tyrosinase.

### 5.3. Increase the Extraction Rate of Tea Effective Ingredients

Tea is popular due to its high polyphenol content. However, the active components in tea are often not completely extracted through traditional water extraction and organic solvent extraction, which may lead to a reduction in the active components of tea and affect its quality [113]. Tannase is an enzyme capable of degrading polyphenols in tea. Polyphenols in tea, such as tannins, combine with other compounds to form precipitates, resulting in reduced extraction efficiency. Through the action of Tannase, these polyphenols can be degraded to release more active components. This method can effectively improve the extraction rate and polyphenol recovery during tea extraction, thus improving the quality of tea beverages [114]. Shao et al. [113] used Tannase to treat tea infusion (70 °C, 40 min). The results showed that the enzymatic reaction significantly increased the total polyphenol content of green tea extract (137–291 g/kg). Compared with water extraction, enzyme extraction effectively degraded ester catechins to non−ester catechins. Wang et al. [114] studied the effect of Tannase immobilization on the biological activity of tea extract. The results showed that after Tannase treatment, the concentration of non−ester catechins EGC and EC increased by 1758% and 807%, respectively, and exhibited high antioxidant activity. Liang et al. [115] studied the effects of different Tannase treatments on theaflavins to develop a tea beverage with potential functions. The results showed that compared with pure withered leaf fermentation, the green tea extract treated with Tannase showed significant effects, increasing theaflavin levels by 4.7 times.

### 5.4. Improve the Flavor of the Tea Infusion

The bitterness and astringency of tea and its extracts make them less favored by consumers, greatly limiting their application in processing. The high content of esterified catechins is one of the key factors contributing to the poor taste of tea infusion. Therefore, effectively reducing the impact of these compounds and improving the taste and quality of tea is a critical aspect of tea processing. Tannase specifically degrades the ester bonds in esterified catechins, converting them into non−esterified catechins, which are primarily responsible for the sweetness, thereby enhancing the taste of the tea infusion [116]. Govindarajan et al. [117] studied the biotransformation of black tea by Tannase derived from *E. cloacae* 41 and conducted a sensory evaluation of the samples. The results showed that in the tea infusion treated with Tannase, the concentrations of EGC, ECG, and gallic acid increased compared to those of the control. Furthermore, the Tannase−treated black tea showed significant improvements in flavor, taste, overall acceptability, and appearance. Summer green tea accounts for about half of the total tea production, but compared to spring green tea, summer tea has a higher content of esterified catechins, which results in a distinct bitterness and astringency, leading to significant waste of summer tea each year. Cai et al. [118] studied the effect of Tannase derived from *A. niger* RAF106 on the flavor of summer tea. The results showed that after Tannase treatment, the contents of non−esterified catechins, soluble sugars, total flavonoids, gallic acid, and free amino acids in summer tea increased significantly by 114.8%, 95.59%, 54.70%, 3775%, and 18.18%, respectively. The fermented tea exhibited a fragrant aroma, a smooth, mellow taste, and a sweet aftertaste, effectively improving the flavor quality of summer tea and laying the foundation for the development of functional tea beverages. Zhang et al. [119] studied the effect of Tannase hydrolysis of EGCG and ECG on the sweetness of green tea infusion. The results showed that with the extension of hydrolysis time, both the sweetness intensity and overall acceptability of the green tea infusion significantly increased. Moreover, when the EGC/EC ratio was 2.5:1 and the total concentration was 3.5 mmol/L, the best sweet aftertaste effect was achieved.

**Figure 3 foods-14-00079-f003:**
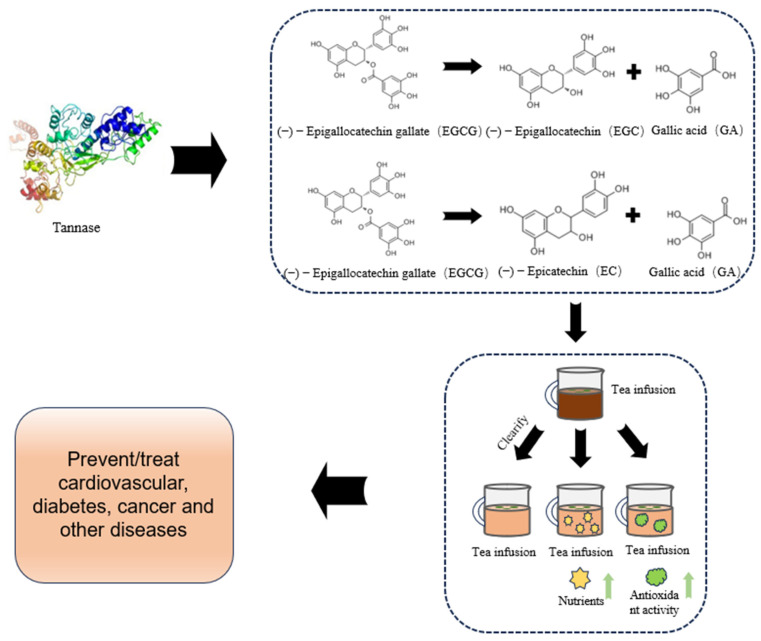
Application of Tannase in tea beverage (Tannase 3D structure prediction from ([120]).

## 6. Summary and Prospect

Tannase has great application potential in the tea industry, but in practical application, due to low expression, high cost, and complex purification and recovery, these factors seriously restrict the wide application and industrial development of Tannase. Therefore, researchers regulate and optimize gene cloning, expression vector construction, and host cell screening through genetic engineering technology, which has a significant effect on improving the expression level of Tannase and reducing the production cost. In addition, this study focuses on the fermentation conditions, enzymatic properties, and purification methods of Tannase. On the one hand, optimizing fermentation conditions can not only improve the yield and purity of Tannase, but also reduce the difficulty and cost of subsequent separation and purification. Moreover, selecting appropriate protein purification methods can improve the purification multiple and recovery rate, reduce production costs, and improve the competitiveness of Tannase. Lay the foundation for large−scale production and industrial application and promote Tannase to continue to advance in the direction of the tea industry. Tannase is widely used in the food industry as a safe and efficient tannin−degrading enzyme, and microbial Tannase has become a research hotspot due to its advantages such as easy access, diversity of properties, and easy genetic engineering modification. 

Remarkable achievements have been made in the past decades, but the following aspects are still worthy of further study: (1) The properties of Tannases from different sources are quite different. If the three−dimensional structure of Tannases from different sources can be studied, the fundamental causes of the property differences can be explained from the molecular level through molecular simulation, site−specific mutation, and other biotechnology, so as to provide the research basis for the construction of excellent Tannases. (2) At present, there have been a lot of studies on gene cloning and recombinant expression through molecular biology and genetic engineering technology, but few reports on optimization in this area. Therefore, further studies can be made on the influence of promoter, signal peptide, intracellular chaperone protein, and other factors on Tannase expression activity. The optimized expression system can be applied in actual production, thereby significantly reducing the preparation cost of Tannase. (3) At present, there are still some shortcomings in Tannase’s high specificity for the complex polyphenol system in tea and its adaptability to the environment during tea processing. In view of this, on the one hand, it is necessary to actively search for Tannase with high specificity to tea. On the other hand, the existing Tannase can be modified with the help of genetic engineering and other technologies, so that it can better adapt to the environment in tea and fully play a role in the complex environment of tea processing.

## Data Availability

No new data were created or analyzed in this study. Data sharing is not applicable to this article.

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
