# Peer review of "Recent Advances of Tannase: Production, Characterization, Purification, and Application in the Tea Industry"

_foods, 2024, doi:10.3390/foods14010079_

Round 1
Reviewer 1 Report
Comments and Suggestions for Authors
In this study, the production, properties, purification, and application of tannase were systematically reviewed, which can provide a reference for further research and application of tannase. However, there are some contents that need to be improved. They are listed as follow:
1. Line 115-116. The four steps for preparing tannase need to be described.
2. Table 3. “Gallic acid was titrated by 0.01N NaOH” need to be modified. What are the substrates, terminators, and color developing agents of this method?
3. Table 4. “Fe3+” should be changed to “Fe3+”.
4. Line 610, “WANG J-Q”. Line 617 “LU M-J“. Line 893, “Lu, M.-J”. The format of authors' names needs to be modified.
5. Line 642, “RAJENDRA F M [83] et al”. Line 665, “WANG C [85] et al”. The position of the reference number is incorrect.
6. Line 859 and 884. The author's name is represented in capital letters.
7. The writing format and unit symbol of the manuscript should to be modified according to the requirements of the journal.
8. The format of the references needs to be reviewed one by one, and some more recent references need to be added.
Author Response
|
In this study, the production, properties, purification, and application of tannase were systematically reviewed, which can provide a reference for further research and application of tannase. However, there are some contents that need to be improved. They are listed as follow: Response: Thank you very much.
Comments 1: Line 115-116. The four steps for preparing tannase need to be described. |
|
Response 1: Thank you for pointing this out. Currently, the preparation of tannase is mainly divided into four steps: sample collection, enrichment culture, primary screening and rescreening. First, suitable samples need to be selected. Tannin-rich samples have great potential for screening tannase-producing strains. Second, a primary screening medium should be prepared, with tannic acid as the sole carbon source. If the strain can grow on the primary screening medium, it indicates the strain's potential to produce tannase. Next, strains capable of producing tannase are fermented and screened, and those with desirable properties are selected. Finally, the selected strains are used for fermentation to extract tannase. We have added detailed descriptions of the four steps involved in tannase preparation in lines 110-166.
|
|
Comments 2: Table 3. “Gallic acid was titrated by 0.01N NaOH” need to be modified. What are the substrates, terminators, and color developing agents of this method? |
|
Response 2: Thank you for pointing that out. Tannase activity is determined by recording the consumption of NaOH. When the NaOH consumption rate stabilizes or approaches zero, it indicates that the reaction has been completed. This method is simple and direct, without the need for chromogenic agents or terminators. Therefore, to maintain consistency in Table 3, the titration method has been removed from the table and a detailed description is provided in Section 3.1.2.
Comments 3: Table 4. “Fe3+” should be changed to “Fe3+”. Response 3: Sorry for the mistake. We have corrected “Fe3+” to “Fe3+” in Table 4.
Comments 4: Line 610, “WANG J-Q”. Line 617 “LU M-J“. Line 893, “Lu, M.-J”. The format of authors' names needs to be modified. Response 4: Sorry for the mistake. We have corrected the formatting of the authors' names as follows: Line 684: “Wang”, Line 688: “Lu”, and Line 1062: “Lu, M.J.”
Comments 5: Line 642, “RAJENDRA F M [83] et al”. Line 665, “WANG C [85] et al”. The position of the reference number is incorrect. Response 5: Sorry for the mistake. We have corrected the reference positions in lines 684 and 688.
Comments 6: Line 859 and 884. The author's name is represented in capital letters. Response 6: Sorry for the mistake. We have corrected them to "Ma, W.L." and "Chen, Z.M." in lines 1021 and 1056, respectively.
Comments 7: The writing format and unit symbol of the manuscript should to be modified according to the requirements of the journal. Response 7: Thank you for pointing this out. We have modified the writing format and unit symbols according to the journal’s requirements and made changes in lines 82, 97, 181, 241, 253, 277, 465, 468, 472, 486, 602, and 631. Additionally, the format of the references in the paper has been unified.
Comments 8: The format of the references needs to be reviewed one by one, and some more recent references need to be added. Response 8: Thank you for pointing this out. We have reviewed the references one by one and added the most recent references: [24], [29], [31], [32], [33], [34], [35], [37], [38], [40], [47], [51], [58], [74], [77], [78], [80], [85], [87], [91], [96], [101], [104], [107], [115], [116], [117], and [118]. |

Reviewer 2 Report
Comments and Suggestions for Authors
The purpose of this review manuscript is to summarize the recent advances of tannase in production, purification, and application in the tea industry, however, the references cited in this manuscript is not state of the art. So, the knowledge described in this MS is not the latest research progress of the tannase production. The current MS is not suitable for publication until significant improvements will be made. Some specific comments are as follows.
1. Some sentences are difficult to be understanded: i.e. Chaitanyakumar A et al. [13] used S. lugdungensis DNA as a template to amplify the expression of the TanA gene in Escherichia coli for the first time, and conducted immobilized studies, which found that the immobilized tannase had protective effects in a wide range of pH and temperature. Compared to other bacterial tannases, tannases are suitable for various industrial applications.
2. Fig. 2. Tannase preparation process (solid state fermentation for example): Is it the industrial or lab-scale process of tannase production?
3. Table 2 Fermentation conditions of tannase: Is it a solid state fermentation or Liquid fermentation?
4. In 4. Purification method of tannase: Many references cited in this section are not specific for tannase purification.
Comments on the Quality of English LanguageEnglish Language needs to be improved.
Author Response
|
The purpose of this review manuscript is to summarize the recent advances of tannase in production, purification, and application in the tea industry, however, the references cited in this manuscript is not state of the art. So, the knowledge described in this MS is not the latest research progress of the tannase production. The current MS is not suitable for publication until significant improvements will be made. Some specific comments are as follows. Response: Thank you very much.
Comments 1: Some sentences are difficult to be understanded: i.e. Chaitanyakumar A et al. [13] used S. lugdungensis DNA as a template to amplify the expression of the TanA gene in Escherichia coli for the first time, and conducted immobilized studies, which found that the immobilized tannase had protective effects in a wide range of pH and temperature. Compared to other bacterial tannases, tannases are suitable for various industrial applications. |
|
Response 1: Sorry for the mistake. We have removed the sentence and cited more recent references as examples, specifically in lines 99-103.
|
|
Comments 2: Fig. 2. Tannase preparation process (solid state fermentation for example): Is it the industrial or lab-scale process of tannase production? |
|
Response 2: Thank you for pointing this out. Fig. 2 refers to the laboratory-scale production of tannase. Currently, the preparation of tannase is mainly divided into four steps: sample collection, enrichment culture, primary screening and rescreening. First, suitable samples need to be selected. Tannin-rich samples have great potential for screening tannase-producing strains. Second, a primary screening medium should be prepared, with tannic acid as the sole carbon source. If the strain can grow on the primary screening medium, it indicates the strain's potential to produce tannase. Next, strains capable of producing tannase are fermented and screened, and those with desirable properties are selected. Finally, the selected strains are used for fermentation to extract tannase. We have added detailed descriptions of the four steps involved in tannase preparation in lines 111-167.
Comments 3: Table 2 Fermentation conditions of tannase: Is it a solid state fermentation or Liquid fermentation? Response 3: Thank you for pointing this out. We have added the fermentation method of tannase in Table 2.
Comments 4: In 4. Purification method of tannase: Many references cited in this section are not specific for tannase purification. Response 4: Thank you for pointing this out. We have added some recent references in Section 4: [77], [78], [79], [80], [85], [87], [95], [96], and [101]. Although some references used to introduce the principles of purification methods may not be specific to tannase, all the examples provided in Section 4 are directly related to the purification of tannase.
|

Reviewer 3 Report
Comments and Suggestions for Authors
This paper presents a comprehensive review of tannase, a tannin-degrading enzyme. Although there are similar papers already published, to my knowledge this is the only paper that focuses not only on characterization but also on purification and extraction methods for tannase production. This paper will presumably be of value to researchers in the same field. However, there are a few observations which can contribute to a better understanding of the Manuscript.
The paper is a collaboration of different authors and that is clear from the text. Therefore, I would suggest the authors to review the whole Manuscript and to uniform the style and the language.
Line 459-460 duplicate text
Line 700 Please rephrase
Please, uniform the references in the text (small letters, all caps…?)
I suggest to add the tannase activity in the Tables e.g. Table 2
Author Response
|
This paper presents a comprehensive review of tannase, a tannin-degrading enzyme. Although there are similar papers already published, to my knowledge this is the only paper that focuses not only on characterization but also on purification and extraction methods for tannase production. This paper will presumably be of value to researchers in the same field. However, there are a few observations which can contribute to a better understanding of the Manuscript. Response: Thank you very much.
Comments 1: The paper is a collaboration of different authors and that is clear from the text. Therefore, I would suggest the authors to review the whole Manuscript and to uniform the style and the language. |
|
Response 1: Thank you for pointing this out. We have carefully reviewed the entire manuscript to ensure consistency in style and language.
|
|
Comments 2: Line 459-460 duplicate text |
|
Response 2: Sorry for the mistake. We have deleted duplicate text at line 519.
Comments 3: Line 700 Please rephrase Response 3: Thank you for pointing this out. We corrected it in line 882 to read " The optimized expression system can be applied in actual production, thereby significantly reducing the preparation cost of tannase."
Comments 4: Please, uniform the references in the text (small letters, all caps…?) Response 4: Thank you for pointing this out. We have unified the format of references in the text.
Comments 5: I suggest to add the tannase activity in the Tables e.g. Table 2 Response 5: Thank you for pointing this out. We have added tannase activity in Table 2. |

Round 2
Reviewer 2 Report
Comments and Suggestions for Authors
The revised manuscript was significantly improved. English still needs to be carefully revised. for example the title "Recent Advances of tannase: Production, characterization, purification, and application in the tea industry" The fisrt letter of Advances should be lowercase.
Comments on the Quality of English Language
needs to be improved
Author Response
- Summary
Thank you very much for taking the time to review this manuscript. We appreciated very much the reviewers’ constructive and insightful comments. In this revision, we have addressed all of these suggestions. We hope the revised manuscript has now met the publication standard of your journal. We highlighted all the revisions in red colour. On the next pages, our point-to-point responses to the queries raised by the reviewers are listed.
- Point-by-point response to Comments and Suggestions for Authors
The revised manuscript was significantly improved. English still needs to be carefully revised. for example the title "Recent Advances of tannase: Production, characterization, purification, and application in the tea industry" The fisrt letter of Advances should be lowercase.
Response: Sorry for the mistake. We have carefully read and revised the manuscript.
